# Porcine Reproductive and Respiratory Syndrome Virus: Immune Escape and Application of Reverse Genetics in Attenuated Live Vaccine Development

**DOI:** 10.3390/vaccines9050480

**Published:** 2021-05-10

**Authors:** Honglei Wang, Yangyang Xu, Wenhai Feng

**Affiliations:** 1State Key Laboratory of Agrobiotechnology, China Agricultural University, Beijing 100193, China; wanghl@cau.edu.cn (H.W.); SZ20183020174@cau.edu.cn (Y.X.); 2Ministry of Agriculture Key Laboratory of Soil Microbiology, China Agricultural University, Beijing 100193, China; 3Department of Microbiology and Immunology, College of Biological Sciences, China Agricultural University, Beijing 100193, China

**Keywords:** PRRSV, immune evasion, vaccine, reverse genetics

## Abstract

Porcine reproductive and respiratory syndrome virus (PRRSV), an RNA virus widely prevalent in pigs, results in significant economic losses worldwide. PRRSV can escape from the host immune response in several processes. Vaccines, including modified live vaccines and inactivated vaccines, are the best available countermeasures against PRRSV infection. However, challenges still exist as the vaccines are not able to induce broad protection. The reason lies in several facts, mainly the variability of PRRSV and the complexity of the interaction between PRRSV and host immune responses, and overcoming these obstacles will require more exploration. Many novel strategies have been proposed to construct more effective vaccines against this evolving and smart virus. In this review, we will describe the mechanisms of how PRRSV induces weak and delayed immune responses, the current vaccines of PRRSV, and the strategies to develop modified live vaccines using reverse genetics systems.

## 1. Introduction

Porcine reproductive and respiratory syndrome (PRRS), first reported in the United States in 1987, is one of the most important porcine diseases characterized by severe reproductive failures in pregnant sows and respiratory disorders in pigs of all ages. The global pig industry has suffered huge economic losses due to the prevalence of PRRS [1]. The causative pathogen, PRRS virus (PRRSV), is an enveloped single-stranded positive RNA virus, belonging to the family of *Arteriviridae* in the order *Nidovirales*. The viral genome is approximately 15.4 kb, which contains 11 known open reading frames (ORFs) [2]. ORF1a and ORF1b, occupying 5′-end three-quarters of the viral genome, encode two large polyproteins, pp1a and pp1ab, which are synthesized from the genome RNA template. The polyproteins are then processed by a complex proteolytic cascade to generate at least 14 nonstructural proteins (nsps): nsp1α, nsp1β, nsp2–6, nsp7α, nsp7β, and nsp8–12 [3]. There are another two novel proteins, nsp2TF and nsp2N, which are expressed through a -2/-1 programmed ribosomal frameshift mechanism [3]. ORF2a, ORF2b, ORFs 3–7, and ORF5a are located at the 3′-end of the viral genome, encoding 8 structure proteins, including glycoprotein (GP) 2, envelope (E), GP3, GP4, GP5, ORF5a, membrane (M), and nucleocapsid (N) protein [4]. In virus particles, the structural proteins participate in the virus internalization process [5].

PRRSV is classified as PRRSV-1 (species Betaarterivirus suid 1) and PRRSV-2 (species Betaarterivirus suid 2) [6], which are identified in 1991 in Europe and 1992 in North America, respectively. PRRSV-1 and PRRSV-2 share about 50–70% nucleotide sequence identity [7]. Based on phylogenetic analysis of ORF5 sequences, which encode the most variable PRRSV structural protein GP5, PRRSV-1 is classified into four subtypes and PRRSV-2 is divided into nine lineages with an inter-lineage genetic distance of 11–18% [8]. The distribution and prevalence of each subtype/lineage are investigated every few years. High-frequency mutation and recombination between different lineages/sub-lineages are important reasons for the diversity of PRRSV and may be of great significance to accelerate PRRSV evolution, which has been evidenced in cellular and animal experiments [7]. For example, genomic mutations are characteristic of the highly pathogenic PRRSV (HP-PRRSV) with a discontinuous 30-amino-acid deletion in nsp2, which appears to be more pathogenic than parental strains and highly transmissible and has been prevalent in China since 2006 [9]. The most recent evolution evidence is the appearance of NADC30-like strains in China, which have the highest nucleotide similarity with NADC30 strains in the United States [10]. It still happens that NADC30-like strains recombine with Chinese traditional and HP-PRRSV strains in the fields. Moreover, recombination between field and vaccine strains has been found [11,12]. The expansion of genetic diversity of PRRSV, accompanying expansion of antigen diversity, presents more challenges for the vaccines in providing comprehensive and effective protections.

At present, vaccination is one of the most effective methods to control PRRS. Modified live vaccines (MLVs) and inactivated virus vaccines are commercially available [13,14,15]. DNA vaccines, subunit vaccines, and vector vaccines have also been studied extensively [14]. Unfortunately, currently available vaccines only provide protection against homologous strains, but unable to provide effective cross-protection to heterologous strains [16]. Besides great genetic and antigenic heterogeneity, immune evasion of PRRSV is also a major obstacle to controlling PRRS. Here, we will review the immune response to PRRSV infection and the efforts to enhance the scope and intensity of vaccine protections.

## 2. Aberrant Immune Responses Induced by PRRSV

PRRSV infection results in viremia within one week and usually lasts for 4–5 weeks. When viremia disappears, PRRSV will be almost undetectable in the bloods and lungs of pigs [17,18]. However, PRRSV is still present in lymphoid tissues, including tonsils and lymph nodes, and persists over several months or “lifelong” [19]. Although the pig’s immune system is capable of eventually eliminating the virus, it takes a long time [20]. The persistence of the virus may be the result of poor immune responses.

### 2.1. Suppression of Type I Interferon (IFN) Production and Signaling

Type I interferons are the most potent antiviral cytokines against virus infection. In the case of RNA virus infection, viral RNA is sensed by RIG-I-like receptors (RLRs) or toll-like receptors (TLRs), and subsequently turns on the signaling cascades, resulting in the activation of transcription factors IRF-3/7 and NF-κB to increase IFN-α/β production [21]. After being secreted, IFN-α/β binds to the receptors IFNAR1 and IFNAR2 to trigger JAK-STAT signaling pathway activation, and then induces the expression of IFN-stimulated genes (ISGs) to maintain an antiviral status in cells [22].

Unfortunately, PRRSV has mechanisms to suppress interferon productions to evade host immune responses. It has been observed that the production of IFN-α is negligible in lung secretions of PRRSV-infected pigs, as well as in PRRSV-infected porcine alveolar macrophages (PAMs) and dendritic cells (DCs) [23]. PRRSV still blocks the production of IFN-α after superinfection with swine transmissible gastroenteritis virus (TGEV), a good IFN-α inducer [24]. Additionally, PRRSV is found to inhibit IFN-α-induced ISG15 and ISG56 productions in PAMs and MARC-145 cells [25]. These data suggest that PRRSV modulates host antiviral defenses by inhibiting IFN responses.

So far, several viral proteins have been identified as IFN antagonists (Figure 1). PRRSV nsp1α and nsp1β have papain-like cysteine protease (PLP) α and PLPβ activities, respectively, which mediate the rapid release of nsp1α and nsp1β [26,27]. nsp1α and nsp1β are reported to be effective in antagonizing IFN responses. PRRSV nsp1α inhibits the activation of NF-κB, thus interfering with the activity of the IFN promoter [28]. In addition, nsp1α degrades CREB binding protein (CBP) via the proteasome-dependent degradation pathway and inhibits the binding of IRF3 and CBP in the nucleus, leading to the blockade of IFN expressions [29]. It is reported that PRRSV nsp1β inhibits the phosphorylation and nuclear translocation of IRF3. Moreover, nsp1β inhibits the nuclear translocation of IFN-stimulated gene factor 3 (ISGF3) through degrading karyopherin-α1 (KPNA1), resulting in the suppression of ISGs expressions [30]. Recently, nsp1β is identified to disintegrate the nuclear pore complex (NPC), thus confining host cell mRNAs in the nucleus and suppressing host protein synthesis. However, the nsp1β mutant loses its ability to interfere with IFN production, IFN signaling, and TNF-α production [31].

PRRSV nsp2, the largest and most variable nonstructural protein, serves as membrane scaffold proteins together with nsp3 and nsp5 to induce the assembly of replication and transcription complexes, and plays an important role in regulating IFN responses [32]. PRRSV nsp2 has PLP2 activity, which belongs to the ovarian tumor (OTU) protease family and possesses deubiquitinating (DUC) and deISGylating activities, thus inhibiting Ub and ISG15-dependent antiviral signals [33]. PRRSV nsp2 exerts its DUB potential to inhibit the NF-κB signaling pathway by preventing the polyubiquitination and degradation of IκBα [34]. PRRSV nsp2 also inhibits IRF3 phosphorylation, which depends on its PLP2 domain [35]. However, nsp2 in some strains cannot inhibit IFN production, probably because nsp2 sequences are different among strains [36].

PRRSV nsp4 is a 3C-like serine protease, responsible for nsp 3–12 processing, and can inhibit IFNβ promoter activity and transcription [37]. Huang et al. [36] show that HP-PRRSV nsp4 impairs NF-κB activation by hydrolyzing NF-κB essential modulator (NEMO) at Glu349. A recent study shows that nsp4 also cleaves NEMO at two other sites, Glu166 and Glu171 [38]. Additionally, nsp4 inhibits the activation of RIG-I signaling pathway by cleaving VISA, thereby down-regulating type I IFN production [39]. It is reported that the inhibitory effect of HP-PRRSV nsp4 on IFNβ is greater than that of the low pathogenic strains [40].

PRRSV nsp11 has endoribonuclease (NendoU) activity at its C-terminal domain, cleaving ssRNA and dsRNA at specific sites of uridylic acid [41]. Nsp11 can inhibit the activity of IFN-β promoter by restricting IRF3 and NF-κB activation [42]. Sun et al. find that nsp11 reduces MAVS and RIG-I expressions, thus suppressing RLR-mediated pathway and IFN-β expression [43]. Nsp11 mutants that have impaired EendoU activity fail to inhibit IFN production [42,43], suggesting that the NendoU activity of nsp11 plays an important role in reducing IFN production. Recently, it is reported that nsp11 interacts with IRF9 to impair the formation and nuclear translocation of IFN-stimulated gene factor 3 (ISGF3) [44], and induces STAT2 degradation to block IFN signaling [45]. However, these activities are independent of NendoU activity.

In addition, N protein inhibits the phosphorylation and nuclear translocation of IRF3 [46]. A recent study shows that N protein inhibits the expression of TRIM25 and TRIM25-mediated RIG-I ubiquitination, thereby down-regulating IFN-I production [47]. PRRSV nsp7 suppresses the expression of IFNs and ISGs [48]. Moreover, nsp2TF and nsp2N are also found to antagonize IFN-α [49].

IFN can act on antigen-presenting cells (APCs) to enhance their ability to activate T cells to drive maturation from a naïve T cell to an effector T cell. The meager IFN response might result in a weak adaptive immune response to PRRSV. Pigs receiving recombinant porcine IFN-α treatment have delayed viremia, decreased viral loads, and an increase in the number of virus-specific IFN-γ-secreting cells [50].

In conclusion, PRRSV can modulate type I IFN production and function via various mechanisms. The approach to increase IFN signaling should be hopeful to develop antiviral responses. It has been reported that a RIG-I stimulating RNA, which possesses dual functions (a siRNA targeting IAV NP gene and an agonist for RIG-I activation), is developed as influenza therapeutics [51]. In addition, microRNAs are critical gene regulators by targeting 3′ UTR of mRNAs at post-transcriptional level, and play important roles in antiviral immune responses. For example, miR-23 and miR-26a can enhance IFN-I and ISGs expressions during PRRSV infection [52,53]. The development of the miRNA preparation and delivery system provides great hope for the therapeutic potential of miRNA-based therapy. Besides, the miRNA-based strategy can be used to develop live attenuated vaccines.

### 2.2. Dysregulation of Cellular Immune Responses

It has been established that PRRSV replication in bone marrow cells and the induction of cell apoptosis results in bone marrow hypoplasia characterized by a lack of normal myeloid and erythroid precursors [54,55]. Amarilla et al. report that the percentage of affected hematopoietic tissue and stroma is independent of viral loads and virulence at the early stages of PRRSV infection [55]. Thus, all PRRSV strains infection may lead to a decrease in progenitor cells and ultimately a decrease in total white blood cells.

Thymus is the main site that supports T cell proliferation, differentiation, and repertoire selection [56]. PRRSV infects thymus cells, and the viral loads and the pathological degree of thymus are related to virus virulence [57,58]. Low pathogenic PRRSV leads to mild thymic atrophy and fewer cell apoptosis [57]. However, HP-PRRSV infection causes severe thymus lesions, including cortical atrophy, smaller thymic lobules, rupture of the boundary between cortex and medulla, extensive apoptosis of thymocytes, thymic epithelial cell autophagy, and decreased CD4^+^CD8^+^ thymocytes [57,59]. Recently, Ruedas-Torres et al. observe a positive correlation between PRRSV-N-protein positive cells and apoptosis-related markers in thymic cortex, and thymocyte apoptosis is likely through exogenous apoptotic pathways, because the number of cells expressing Caspase 8 and Fas in the cortex and medulla of both high and low pathogenic PRRSV-1 infected piglets are increased [59]. The apoptosis of thymocytes might result in a decrease in the number of mature T lymphocytes [60]. This may explain why more severe clinical signs are observed in HP-PRRSV-infected pigs.

PRRSV infection also leads to lesions in secondary lymphoid tissues, including spleen, lymph nodes (LN), tonsils, and mucosal-associated lymphoid tissues [61,62]. Apoptotic cells are detected in B- and T-cell areas of lymphoid organs [63].

PRRSV infection causes a delayed induction of effector T cells in peripheral blood [64,65]. Of the peripheral blood mononuclear cells (PBMC) in PRRSV infected pigs, Th1, Th17, and cytotoxic T lymphocytes (CTLs) responses are induced 21–35 days post challenge (dpc), when most of the virus has been cleared from the blood [17]. However, early and effective cellular immune responses are triggered in local tissues. Th1, Th17, and CTLs are significantly increased at 10 dpc in lymphoid tissues and lung parenchyma of PRRSV infected pigs, when the virus levels are at the peak in the body [17]. Another report also shows that local T-cell responses are maintained at high levels at 10 dpc [66], implying that there is a critical role played by local immune responses in the clearance of PRRSV.

Tregs are known to possess immunosuppressive properties, which are able to inhibit APCs and the activation and proliferation of CD4^+^ and CD8^+^ T cells [67]. However, in PRRSV-infected animals, Treg response is controversial. Several studies have revealed that Tregs are increased after PRRSV infection [68,69]. In contrast, other studies reveal that Tregs are not changed or even suppressed by PRRSV infection [17,65]. The conflict results may be due to the difference of strains or the method used to evaluate Treg reaction. Thus, it is speculated that Tregs may only play a minor role during acute infections [17].

There are no notable changes in γδ T cells at 28 days post-PRRSV infection. However, following re-exposure to HP-PRRSV, γδ T cells significantly increase in proliferation and IFN-γ production [70]. Kick et al. suggest that these γδ T cells play important roles in the immune response to PRRSV in lymphoid tissues [70]. The role of γδ T cells is still unclear, and further exploration is required to determine whether γδ T cells exacerbate or eliminate PRRSV infection.

Dendritic cells (DCs) are the most potent APCs that play an important role in T cell activation [71]. Some viruses, such as PRRSV, have evolved to specifically target DCs to evade host immune responses. PRRSV infection induces DCs death via apoptosis and necrosis mechanisms [72]. PRRSV infection reduces the expression of MHC-I/MHC-II molecules in monocyte-derived dendritic cells (Mo-DCs) and bone marrow-derived dendritic cells (BMDCs) [73,74,75], which are consistent with the impairment of T cell activation in PRRSV-infected pigs. However, a recent study has shown that PRRSV infection induces functional up-regulation of the swine leukocyte antigen (SLA)-DR in BMDCs while inhibits SLA-DR at mRNA level [76]. The inconsistency attributes to the inhibition of the ubiquitin-mediated degradation of SLA-DR by the PRRSV nsp2 OTU domain. The up-regulated SLA-DR triggers non-neutralizing antibody responses in the early time of viral infection that cannot inhibit virus replication. In addition, PRRSV infection can down-regulate the expression of CD80/86 costimulatory molecules and the secretion of IFN-α/β, IL-12, IFN-γ, and TNF-α in DCs, resulting in Th1/Th2 balance unsteadiness [74,77,78]. The state of DC maturation, the cytokine microenvironment, and the antigen presentation ability are aberrant during PRRSV infection, leading to the failure in the onset of efficient T cells immunity.

To improve the cellular immune responses, dendritic cells should be deeply concerned. It has been reported that by delivering a recombinant PRRSV antigen through a recombinant mouse-porcine chimeric antibody specific to the porcine DC-SIGN neck domain, porcine DCs can rapidly internalize them and induce higher numbers of IFN-γ producing CD4^+^ T cells [79]. Furthermore, a plasmid DNA vaccine encoding DC-specific surface molecule DEC-205 single-chain fragment linked with the hepatitis B surface antigen induces robust antiviral T cell and antibody immunity [80]. These data suggest that the recombination of antigen subunit and DC-specific target molecules can be used as a strategy to improve protective immunity against PRRSV by inducing efficient T cell responses. In addition, we need to deeply investigate how PRRSV modulates the function of DCs and find the related virulence factors in PRRSV. Then, we can manipulate the virus to generate recombinant PRRSVs to reduce their damages to DCs, subsequently improving T cell responses to PRRSV.

### 2.3. Antibody Response

The antigenic epitopes are present in many nonstructural proteins (nsp1, nsp2, nsp4, and nsp7) and structural proteins (GP5, M, and N) [81,82]. During PRRSV infection, antibody responses appear rapidly and persist for several months. However, early antibodies have nothing to do with protection, but only for diagnosis [64].

When antibodies binding to virions are insufficient to mediate neutralization, antibody dependence enhancement (ADE) will occur, thereby facilitating the entry of viruses that replicate in FcR-expressing cells such as monocytes and macrophages through FcR [83]. In addition to increasing viral replication and burdens, the viruses can trigger an increased release of cytokines, leading to more severe symptoms. It has been reported that the non-neutralizing antibodies against PRRSV induce the phenomenon of ADE to exacerbate PRRSV infection. Yoon et al. first describe the increase in the level of viremia and viral shedding in vivo after passive transfer of sub-neutralizing antibodies [84]. Lopez et al. also observe that serum transfer of salt-concentrated non-neutralizing IgGs collected at 21 days post-PRRSV infection causes increased interstitial pneumonia [85]. However, they could not rule out the possible role of pro-inflammatory cytokines co-salted out with IgG. A recent article has shown that the exacerbated disease is not caused by anti-PRRSV antibodies induced by an inactivated PRRSV vaccine [86]. In brief, the role of ADE in the pathogenesis of PRRSV has been questioned.

Neutralizing antibodies (NAs) are critical weapons against viruses, and the mechanism by which many effective vaccines work to protect against viral infection is to induce NAs. PRRSV is characterized by delayed and limited effective production of NAs, which are detected at around 28–42 days post-infection when the viremia disappears [87]. Delayed NAs do not prevent the establishment of chronic infection following natural wild-type PRRSV infection, which persists in tissues for months even after the appearance of NAs [88]. However, prophylactic anti-PRRSV NAs have been shown to be effective in protection against subsequent exposure. The passive transfer assay displays that NAs derived from PRRSV-challenged pigs completely prevent transplacental infection of piglets and pregnant sow infection [89]. Sterilizing immunity in piglets requires higher concentrations of NAs transferred (1:32). Relatively low levels of NAs (1:8) can block viremia in piglets but not peripheral tissue seeding and transmission to contact animals, while lower levels of NAs confer sterile immunity against PRRSV in sows [85]. These data indicate that NAs could prevent clinical PRRS, and the titer of NAs is helpful a protective parameter in the evaluation of PRRSV vaccines.

The main targets for PRRSV neutralization are still controversial. Generally, the major neutralizing epitope is considered to be present on GP5 that evokes primarily strain-specific NA responses, and the core sequence of the epitope is mapped to amino acids 37–45 in GP5 [82]. The envelope protein M that forms a dimer with GP5 also induces NA responses. A single amino acid (Tyr 10) deletion in M protein confers resistance to polyclonal antibodies with extensive neutralizing activity [90]. Recently, more reports have shown that minor glycoproteins GP2, GP3, and GP4 contain neutralizing antibody epitopes [91,92]. A study shows that ORF1a contains a NA region [93]. Passive protection of NAs is demonstrated to be effective under homologous conditions. Moreover, several references show that some adult pigs are capable of producing broadly neutralizing antibodies against genetically diverse PRRSV strains [94]. However, the targets of broadly neutralizing antibodies are unclear. These envelope proteins would be candidates for vaccine antigens that can induce broadly neutralizing antibodies, which may be of significance in cross-protection. A recent study reports that a broadly neutralizing monoclonal antibody appears to recognize a specific virus epitope that requires post-translational modification within the host cellular Golgi apparatus [95]. It suggests the existence of a novel epitope that can confer cross-protection. However, our cognition of neutralizing antibodies against PRRSV is insufficient.

## 3. Research Status of PRRS Vaccines

Vaccination is the most effective and practical way to prevent and control infectious diseases. Currently available commercial vaccines against PRRSV infection are inactivated vaccines and modified live virus vaccines (MLVs). These vaccines can effectively reduce clinical diseases and viremia in pigs. However, they are effective mainly against homologous infections rather than heterologous infections. Thus, different strategies should be tried to develop vaccines that induce better immunity and broader protection.

Inactivated vaccines have better safety, but confer very limited protective efficacy. Inactivated vaccines fail to prevent viremia in young pigs and reproductive failure or vertical transmission to their offspring in sows after wild-type PRRSV strain challenges. That is partly because of the lack of PRRSV-specific NAs production and cell-mediated immunity (CMI). As a result, inactivated vaccines have not been available in the United States since 2005 [13]. Over the past years, adjuvant technology has been revolutionized and proper selection of adjuvants is helpful to enhance immune responses [96]. It has been reported that recombinant B subunit of *E. coli* heat-labile enterotoxin (rLTB) and IFN-α as adjuvants for inactivated vaccines can enhance humoral and cellular immune responses [97,98]. In addition, a nanoparticle-entrapped inactivated vaccine with poly(lactic-co-glycolic) acid as an adjuvant could elicit cross-protective immune responses [99]. The data suggest that inactivated vaccines with novel adjuvants are one of the promising approaches to enhance potent PRRSV-specific antibody and cell responses.

Modified live vaccines (MLVs) are considered to be more valuable for PRRS control. Many commercial PRRSV-derived vaccines such as Ingelvac PRRS MLV, CH-1R, JXA1-R, HuN4-F112, and TJM-F92 are developed through serial passages of field PRRSV strains on Marc-145 cells or other cells lines. The strategy causes random mutations and deletions in viral genomes by multiple extensive passages. For example, JXA1-R is obtained by passaging the HP-PRRSV strain JXA1 for 80 passages and there are 45 amino acids changes observed in JXA1-R compared with the parental strain [100]. The attenuated vaccines are effective to reduce clinical disease, viremia, and viral shedding caused by lethal PRRSV challenge [101,102]. However, MLVs cannot provide extensive and adequate protection against different PRRSV strains. The outbreaks of PRRS in China caused by NADC-30 like strains in vaccinated pigs indicate the inefficacy of commercial PRRSV vaccines [103]. Evolving viruses challenge the development of vaccines, and more approaches should be explored to overcome the problem, which will be described in more details below.

Besides inactivated vaccines and MLVs, genetically engineered vaccines, including vector vaccines, subunit vaccines, and DNA vaccines, have been developed with the advantages of simple design and good safety. The vaccines that express PRRSV structural proteins in viral, bacterial, fungal, plant, or DNA vectors have been designed and assessed in pigs or mice. A recombinant TGEV expressing modified GP5 and M proteins is constructed, but it could only provide partial protection against PRRSV [104]. Attenuated pseudorabies virus (PRV) is also evaluated as potential replicating vectors for PRRSV. Pigs immunized with PRV expressing GP5 and modified M proteins have reduced viremia period, viral loads, and lung lesions [105]. Recombinant adenovirus vector expressing GP3, GP5, and porcine GM-CSF fusion proteins is able to induce higher levels of NAs, and pigs immunized with the vaccine have reduced clinical signs, viremia, and lung lesions upon PRRSV challenges [106]. In addition, the baculovirus expression system has been widely used as gene delivery and vaccine development tools for its high transgene capacity. Modified baculovirus expressing GP5 and M proteins can boost anti-PRRSV antibody response and IFN-γ production in vaccinated pigs [107,108]. Some viruses without replication competency are also reported. Replication-defective adenovirus expressing GP3-GP4-GP5 fusion proteins and recombinant modified vaccinia virus Ankara (MVA) expressing GP5-M fusion proteins have been developed as vaccines against PRRSV in a mouse model [109]. Mycobacterium bovis BCG and Kluyveromyces lactis are also used to express GP5/GP5-M proteins. They are reported to induce anti-PRRSV IgG and IFN-γ production in mice [14,110]. Unfortunately, these vaccines have not been tested in pigs. Furthermore, pigs fed with transgenic plants (e.g., banana, potato, or tobacco expressing GP5; corn calli or soybean expressing M and N protein; and Arabidopsis expressing GP3-GP4-GP5 or codon-optimized and transmembrane-deleted recombinant GP4-GP5) can develop PRRSV-specific antibody and cell-mediated immunity [111]. DNA vaccines are plasmids encoding genes of interests. The DNA vaccines of PRRSV expressing GP3, GP5, or M protein have been assessed. Pigs or mice inoculated with the vaccines could elicit anti-PRRSV responses. To enhance the immunogenicity of these vaccines, adjuvants are used. Cytokines (e.g., IL-2, IL-4, IL-12, IL-18, IFN-α/γ, and IFN-λ1), CTLA-4, or porcine glutathione peroxidase-1 (GPX1) as immunomodulators are co-expressed with GP5 or GP3/GP5 or M protein to construct DNA vaccines [14]. Immunization with these DNA vaccines induces less clinical disease and viremia and stronger cell and antibody-mediated immune responses in pigs than their parental DNA vaccines when pigs are challenged with homologous viruses [13,14]. Furthermore, a GP5-Mosaic DNA vaccine can induce some degree of cross-protective immunity [112].

The combined immunization of two vaccines seems to work better. For example, the protective effect of a CTLA4-GP5 based DNA vaccine is further enhanced by a booster immunization with inactivated vaccines [113]. DNA vaccines encoding truncated N protein or encoding B and T epitope antigens derived from PRRSV-1 and MLVs prime-boost regimen induce higher T-cell responses and antibody responses during the challenge experiments [114,115]. It stands as a promising vaccination strategy to improve the control of PRRSV.

## 4. Reverse Genetics for PRRSV MLVs Development

Reverse genetics is the functional analysis of genes by examining the phenotypes of targeted gene changes [116], and has been used to explore various aspects of virus infection, including replication, virulence, pathogenesis, immune responses, vaccine development, and antiviral screening tests [117]. For positive-strand RNA viruses, their genomes are infectious. Upon entry and uncoating, the genomic RNA is immediately translated by host ribosomes to generate one or more polyproteins, which are cleaved by proteases to generate viral proteins. It rarely needs to wrap additional replication-associated proteins in virions for positive-sense RNA viruses.

Now, reverse genetics has been extensively utilized for PRRSV study. The first PRRSV infectious clone, named pABV437, was developed in 1998 from the European strain Lelystad virus [118]. Subsequently, many infectious clones, including classical American strains and high pathogenic PRRSVs, have been constructed. The approaches to construct infectious clones of PRRSV have been described in detail somewhere else [119]. In brief, the cDNA carrying the full-length viral genome is cloned into the downstream of a phage-derived T7 or SP6 promoter prior to RNA transcription in vitro, and the transcribed RNA is then transfected into cells to initiate infection cycles. In another strategy, the entire viral cDNA is cloned into the downstream of a eukaryotic promoter such as the cytomegalovirus (CMV) promoter. The cDNA clone is transfected into cells, and viral genomic RNA is transcribed in the transfected cells to initiate an infection cycle (Figure 2). Mutations can be easily engineered into the viral cDNA genome to obtain mutant viral progenies. The introduction of mutations in viral genomes has provided ideas for the development of live attenuated vaccines, which have been reviewed previously [120]. For example, the attenuated HP-PRRSV constructed by introducing the GFP gene into viral genomic cDNA causes less severe clinical diseases and viremia in pigs [121]. Now, an increasing number of mutations are introduced into the PRRSV genome. Below we will describe the new strategies that have been employed to develop attenuated PRRSV vaccines using reverse genetics systems (Table 1).

### 4.1. Increasing the Production of Type I IFN

PRRSV encodes several proteins involved in innate immune evasion as stated above. Thus, deletion or mutation of the genes involved in immune modulations is one of the methods used to generate novel vaccine candidates with a more attenuated profile or optimized immunogenicity. It is reported that the mutant virus 16-5A (aa 16–20 in nsp1β are substituted with alanines) exhibits reduced virus growth and increased IFN-I expression in vitro. In vivo, it grows much slower at the early stage, but then restores to the level similar to the wild-type, and site mutation occurs, indicating that there is a selection pressure to maintain the IFN-inhibitory property [122]. Similarly, Ke et al. report that two recombinant viruses vL126A and vL135A obtained by mutations at residues L126 and L135 in nsp1β induce mild clinical signs, low viral titers, short duration of viremia, and high levels of IFN-α and NA titers in infected pigs. However, reversion to wild-type sequence is observed in some pigs, and the revertants regain the function to suppress IFN production [123]. Li and colleagues have found that mutation R128A or R129A of a highly conserved motif (123GKYLQRRLQ131) of nsp1β reduces its ability to suppress IFN-I. Then they construct three recombinant viruses (vR128A, vR129A, and vRR129AA) with single or double mutations, and find that the recombinant viruses grow much slower than their parental wild-type virus. The viremia levels in pigs infected with mutant viruses are lower, and IFN-α production increases in the lungs at an early time post-infection, resulting in an increase of NK cells and IFN-γ production [124]. Conversely, the recombinant virus with aa 185 mutation (D185N) in nsp4 replicates more slowly and shows higher abilities to induce IFN-I expression compared with wild-type HP-PRRSV [125]. The virus with the K59A mutation in nsp11 almost loses its ability to suppress STAT2 [45]. Unfortunately, these viruses have not been tested in vivo. Moreover, two recombinant viruses with the knock-out of nsp2TF and nsp2N expedite IFN-α response, increase NK cell cytotoxicity, and enhance T cell immune responses in infected pigs [49]. Whether the attenuated viruses have the potential to be a vaccine needs more study.

PRRSV-A2MC2 has been reported to have a potent ability to induce IFN production. A2MC2 infection results in an earlier onset and higher levels of neutralizing antibodies against both homologous and heterologous strains. This suggests that the enhancement of IFN is a promising method for the production of an effective PRRSV vaccine. However, according to the studies, recombinant viruses with a single nucleotide mutation often have the possibility of restoring virulence [123], and a vaccine that can exist stably is urgently needed. With the deepening of research on the mechanism of PRRSV inhibiting IFN response, more amino acids in PRRSV genome have been identified to modulate IFN signaling. Thus, the combination of multiple mutated viral proteins, including nsp1α, nsp1β, nsp2, nsp4, nsp11, etc., would be a promising strategy for the development of an effective PRRSV vaccine.

### 4.2. Cytokine Adjuvants

Cytokines are often used as adjuvants to improve the immunogenicity of vaccines, since they play an important role in initiating and forming immune responses. In order to induce strong immune responses, some cytokine expression genes have been inserted into the live attenuated PRRSV infectious clones to obtain recombinant viruses using reverse genetic techniques. For example, recombinant PRRSVs (CH-1R strain) expressing porcine IL-4 or porcine granulocyte-macrophage colony-stimulating factor (GM-CSF) are rescued, and remain genetically stable in cell cultures. The two recombinant viruses can induce an increased proportion of CD4^+^CD8^+^ double-positive T cells in pigs post challenge with HP-PRRSV, and the recombinant virus expressing GM-CSF induces higher levels of IFN-γ and lower viremia. However, pigs inoculated with the viruses produce similar humoral responses to that elicited by parental virus CH-1R. In addition, the recombinant virus expressing IL-4 induces a similar protective efficacy to CH-1R [126,127]. Cao and colleagues use IL-15 or IL-18 as an adjuvant by incorporating their encoding genes fused with a membrane-targeting signal into a PRRSV Suvaxyn MLV infectious clone. Although the expression of membrane-bound IL-15 significantly enhances NK cell and IFN-γ-producing cell responses, pigs vaccinated with the recombinant viruses and parental Suvaxyn MLV have similar levels of lung lesions and viral RNA loads after challenge with the heterologous strain NADC20 [128]. These findings indicate that the approach of using cytokines as an adjuvant to enhance cell-mediated immune responses has some effects, and more effective cytokines still need to be explored.

A perfect cytokine adjuvant should activate effective humoral and cell-mediated immune response to enhance the protective efficacy of vaccines. The combination of two or more adjuvants in a vaccine preparation may lead to an improved efficacy. Indeed, several adjuvant combinations have been tested against infectious diseases and cancers [146]. The strategy could be used to design PRRSV vaccines. For example, GM-CSF that can induce an increased cell-mediated immune response could be combined with IL-5, which is important for B cell propagation and differentiation.

### 4.3. Chimeras and DNA Shuffling

Limited or lack of cross-protection is one of the main drawbacks of current vaccines. To broaden the cross-protective range, developing a chimeric vaccine that includes multiple neutralizing epitopes from genetically diverse PRRSV strains seems to be promising. Early studies reveal that chimeric viruses constructed by replacing ORFs 3–6 in VR2332 with the corresponding genes of JA142 [130], or mutating ORFs 2–7 in FL12 strain to the corresponding sequence in LMY [129], have cross-reactive neutralizing antibody responses. Similarly, three chimeric viruses (JAP5, JAP56, and JAP2–6) are obtained by using a VR2332 cDNA infectious clone in which ORF5, ORFs 5–6, and ORFs 2–6 are replaced with the same region obtained from JA142, respectively. The authors inoculate pigs with these three chimeric viruses prior to challenge with VR2332 and JA142 strains and find that all vaccinated pigs have a lower level of viremia than the challenge control pigs. Especially, JAP56 decreases viremia to nearly undetectable levels in pigs challenged with JA142 or VR2332 [131]. The constructed chimeric virus through exchanging nsp2 and GP5-M between HP-PRRSV JXwn06 and LP-PRRSV HB-1/3.9 also induces increased cross-neutralization reactivity [132]. Chimeras between field isolates and MLV strains have been described. Chimeras in which MN184 ORF1 and ORF5–6 replace the genes of Ingelvac PRRS MLV can prevent lung consolidation and attenuate clinical signs [133].

To confer broader ranges of cross-protection to various PRRSV strains, chimeras between more than two strains are also studied. Shabir et al. construct a chimeric virus in which ORF3–4 and ORF5–6 of FL12 are replaced with the genes of two Korean field isolates K08-1054 and K07-2273, respectively. Pigs vaccinated with the chimeric virus have higher levels of TNF-α, IFN-γ, IL-12, and serum virus-neutralizing antibodies, and lower levels of IL-10 and viral loads in the lung. The chimeric virus exhibits better protection levels against K08-1054 and K07-2273 [135].

DNA shuffling is a technology for in vitro recombination of a single gene or pools of homologous genes, which can broaden the antigenic cross-reactivity. The technology is also used to construct chimeras to attenuate PRRSV. For instance, the GP3, GP4, GP5, and M genes of VR2385 or Fostera PRRS MLV strain are individually and entirely shuffled from different parental viruses (VR2385, VR2430, MN184B, JXA1, FL-12, and NADC20) to construct chimeric viruses. The viruses are verified to have broadened cross-neutralizing antibody activities against heterologous PRRSV strains [136,137,138,139,140].

In addition, a previous study analyzes 59 non-redundant complete genome sequences of PRRSV-2 and selects the “concentrated” sequence (most common nucleotide found at each position of the alignment) to generate a shared genome. PRRSV-CON, the recombinant virus produced by reverse genetics, confers significantly broader levels of heterologous protections than wild-type PRRSV [141]. It may be significant for the development of vaccines against genetically variable viruses.

### 4.4. Codon Pair De-Optimization

Most amino acids are encoded by multiple synonymous codons, but the distribution of synonymous codons in different-organism genes is non-random [147]. Different species have different codon preferences. In addition, nucleotides of two adjacent codons are also distributed in a non-random manner. Therefore, utilization of codon pairs in protein-coding sequences is different from the frequency expected in organisms, which is called codon pair bias (CPB) [148]. The codon pair usage is demonstrated to influence protein synthesis efficiency [149]. Virus replication relies on host cell machinery, and the codon pair bias of viral genome usually matches its host. Thus, de-optimization of codon pairs in virulence-related genes has been considered as a strategy to attenuate viruses. In fact, attenuated influenza virus and poliovirus containing underrepresented codon pairs have been constructed [150,151].

The studies of PRRSV have also shown the importance of codon pair bias for its replication and pathogenesis. To attenuate PRRSV, Ni and colleagues de-optimize codon pairs in the GP5 gene by a specific computer program and rescue the codon-pair de-optimized virus. The virus has reduced replicative capacity, and causes lower viremia levels and reduced histological lung lesions [142]. In addition, vaccination of pigs with the attenuated viruses could protect pigs from homologous PRRSV challenges [143]. Gao and colleagues take a similar approach to shuffle the synonymous codons in nsp9 without altering amino acid sequences. They also observe significant reductions in the replication capacity of the acquired virus. The recombination virus could protect pigs from homologous and heterologous PRRSV challenges. All immunized pigs survive without distinct clinical signs and pathological damages. This study also shows that the NA titer and the level of specific IFN-γ expressing CD8^+^ T cells are significantly elevated in vaccinated pigs [144]. Recently, Park et al. obtain similar results with attenuated PRRSV with de-optimization of codon pairs in nsp1. The clinical symptoms of pigs inoculated with the attenuated virus are significantly reduced when a heterologous PRRSV is challenged [145].

Codon pair de-optimization, as a general strategy for vaccine development, has advantages in safety. The codon pair de-optimized viruses contain hundreds of nucleotide mutations. Therefore, there is little likelihood of reversion to virulent forms with even a few point mutations, and the codon pair de-optimized viruses can be easily distinguished from wild-type strains by PCR. Importantly, the marked sequence divergence between codon pair de-optimized viruses and circulating strains also reduce the recombination frequency. While the mechanisms of attenuation are still debated, codon pair de-optimization is a promising way to attenuate viruses. At present, there is no codon pair de-optimized PRRSV vaccine available and much more work is required to be done.

## 5. Concluding Remarks

PRRS is one of the most important diseases in the global swine industry. The need for interventions to blunt the impact of PRRSV has stimulated great progress in our understanding of PRRSV pathogenesis and immunity. Several PRRSV vaccines, especially MLVs have been developed and commercialized to control PRRS. However, the repeated outbreaks of PRRS and the emergence of new PRRSV variants indicate that current vaccines are not fully effective. Indeed, many important questions remain, including cross-protection and safety. The development of an effective PRRSV vaccine with genetic stability against various antigenic variants is urgently required. A combination of several strategies may work to construct an effective PRRSV vaccine. For instance, codon de-optimization strategy that provides for safer live vaccines (less chance of reversion to virulence) is combined with the PRRSV-CON approach that appears to provide broader protection from heterologous virus strains. This approach might provide a path forward for PRRSV vaccine research. With more detailed studies on viral biology, immune responses, and vaccinology, it is hopeful to design an effective, safe, and stable vaccine to prevent PRRSV infection and transmission.

## Figures and Tables

**Figure 1 vaccines-09-00480-f001:**
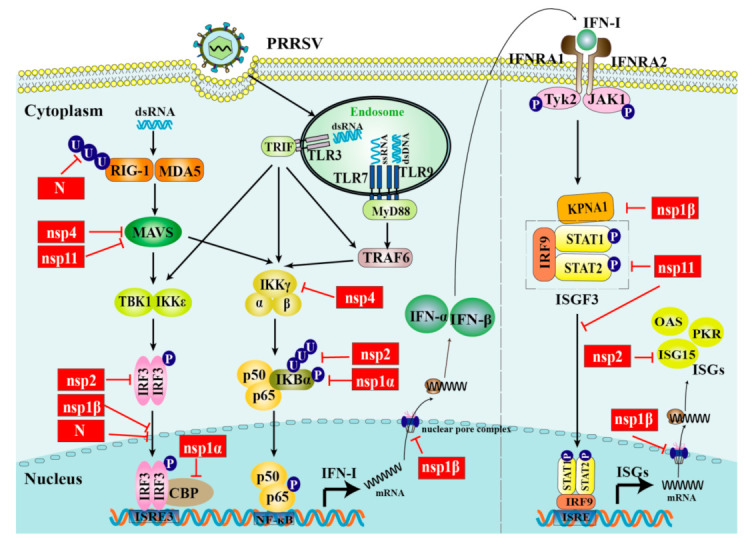
The modulation of the type I interferon (IFN) signal pathway by porcine reproductive and respiratory syndrome virus (PRRSV) proteins. RIG-I and MDA5 detect dsRNA and initiate the adaptor protein MAVS to trigger IRF3 and NF-κB activations. TLR3, located at endosomes, senses dsRNA and transduces signals through TRIF, and TLR7 senses ssRNA and transduces signals through MyD88, leading to the activation of IRFs and NF-κB. After transferring to the nucleus, IRFs and NF-κB will transactivate the promoters to induce type I IFN expressions. IFN-I binds to IFNAR1 and IFNAR2 heterodimers, which transduce signals through recruiting JAK1 and TYK2, and result in STAT1 and STAT2 activation and binding to IRF9, constituting the ISGF3 complex. The complex translocates into the nucleus and promotes ISGs gene expression. PRRSV proteins can hijack multiple steps in type I IFN signal pathways. PRRSV nsp1α inhibits the association between IRF3 and CREB binding protein (CBP), enhances CBP degradation, and interferes with IκB degradation. PRRSV nsp1β inhibits the phosphorylation and nuclear translocation of IRF3, and degrades KPNA1 to block ISGF3 nuclear translocation. PRRSV nsp2 inhibits IRF3 phosphorylation, interferes with IκB polyubiquitination, prevents IκB degradation, and inhibits the ISG15 pathway. PRRSV nsp4 cleaves MAVS/VISA and IKKγ/NEMO. PRRSV nsp11 degrades MAVS mRNA, induces STAT2 degradation to interfere with the formation of ISGF3, and inhibits nuclear translocation of ISGF3. N protein inhibits IRF3 phosphorylation and nuclear translocation, and interferes with TRIM25-mediated RIG-I ubiquitination. P, phosphate; U, ubiquitin.

**Figure 2 vaccines-09-00480-f002:**
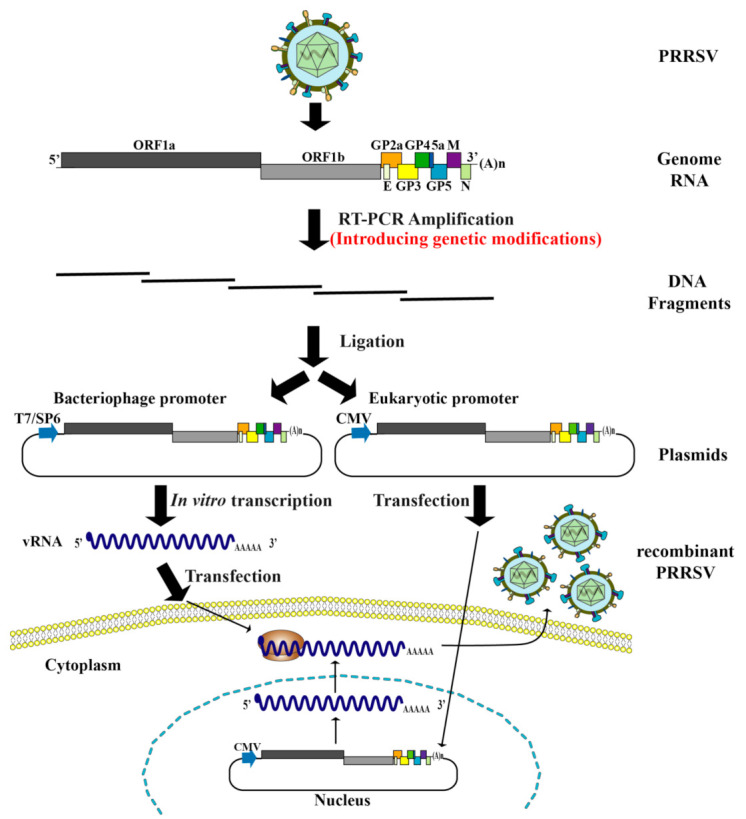
Schematic diagram of the construction of recombinant PRRSVs based on the reverse genetic system. PRRSV genomic RNA is extracted and reverse transcribed to produce cDNA. Several overlapping fragments amplified from the PRRSV genome are ligated with each other and a linearized bacterial plasmid according to homologous recombination technology. During the fragment amplification process, mutations are introduced to generate the full-length infectious cDNA clones with genetic modification. The cDNA cloned downstream of a phage-derived T7 or SP6 promoter is transcribed in vitro, and the transcribed RNA is transfected into cells to initiate infection cycles; or the cDNA cloned into the downstream of a eukaryotic promoter (CMV) is transfected into cells to rescue the recombinant virus.

**Table 1 vaccines-09-00480-t001:** Application of reverse genetics in the development of PRRSV modified live vaccines (MLVs).

Infectious Clone	Modification	Evaluation and References
FL12	nsp1β: 16-5A (aa 16–20 are substituted with alanines)	The mutant virus grows slowly and induces increased IFN-I expression in vitro, but regains wild type growth properties in vivo due to site mutation [122]
FL13	nsp1β: L126A/ L135A	The two recombinant viruses vL126A and vL135A cause mild clinical signs with low viral titers and short duration of viremia, and induce high levels of IFN-α and neutralizing antibody titers in infected pigs. However, reversion to wild-type sequence is observed [123]
SD95-21	nsp1β: R128A/R129A/RR128129AA	The recombinant viruses vR128A, vR129A, and vRR129AA grow more slowly, induce lower levels of viremia, and increase IFN-α production in lungs, resulting in an increase in NK cells and IFN-γ production [124]
HV	nsp4: D185N	The recombinant virus exhibits slowly replication rate and higher ability to induce IFN-I expression in porcine alveolar macrophages [125]
VR2385	nsp11: K59A	The mutant virus almost loses the ability to reduce STAT2 [45]
SD95-21	Δnsp2TF/Δnsp2TFΔnsp2N	The two mutant viruses enhance IFN-α response, NK cell cytotoxicity, and T cell immune responses in infected pigs [49]
CH-1R	Porcine IL-4 gene is inserted between N and 3′-UTR sequence	The virus induces a higher level of IL-4 and proportion of CD4^+^CD8^+^ T cells. But viral load and histopathology do not show significant difference with the parent virus in immunized pigs [126]
CH-1R	Porcine GM-CSF gene is inserted between N and 3′-UTR sequence	The recombinant virus induces a similar humoral response to the parental virus, but a higher proportion of CD4^+^CD8^+^ T cells and IFN-γ level, and lower viremia [127]
Suvaxyn MLV	Porcine IL-15 gene with a membrane targeting signal is inserted to ORF1b/2 junction region	The virus significantly enhances NK cell response and IFN-γ-producing CD4^−^ CD8^+^ T cells and γδ T cells. Pigs vaccinated with the recombinant virus have reduced lung lesions and viral loads after heterologous challenge with PRRSV NADC20 [128]
FL12	ORFs 2–7: replaced by the corresponding sequence of LMY	The chimeric virus has a cross-reactive neutralizing antibody response [129]
VR2332	ORF5/ORFs 5–6/ORFs 2–6/ORFs 3–6: replaced by the same regions of JA142	The substitution reverses the susceptibility of the virus to neutralization antibodies [130]. The viruses (JAP5, JAP56, and JAP2–6) decrease viremia in inoculated pigs challenged with VR2332 and JA142 [131]
JXwn06	nsp2 and GP5-M: replaced by the same regions of HB-1/3.9	The virus induces increased cross-neutralization reactivity [132]
Ingelvac PRRS MLV	ORF1/ORFs 5–6: replaced by the same regions of MN184	The chimeras attenuate the clinical symptoms of infected pigs [133,134]
FL12	ORFs3–4 and ORFs5–6: replaced by the corresponding sequence of K08-1054 and K07-2273, respectively	Viral loads in chimeric virus infected-lungs are low. The chimeric virus induces high levels of TNF-α, IFN-γ, IL-12, and virus-neutralizing antibodies, and low levels of IL-10, and exhibits better protection levels against K08-1054 and K07-2273 [135]
VR2385	ORF3/ORF4/ORF5/ORF6/ORFs 3–6: shuffled ORFs sequence from VR2385, VR2430, MN184b, FL-12, JXA1, and NADC20	Pigs inoculated with the viruses have reduced viral loads, fewer lung lesions, and high levels of cross-neutralizing antibodies against heterologous strains [136,137,138,139]
Fostera PRRS MLV	ORFs 2–6: shuffled ORFs sequence from VR2385, VR2430, MN184b, FL-12, JXA1, and NADC20	The viruses induce cross-neutralizing antibodies against heterologous strains [140]
PRRSV-2	Full-genome: “concentrated” sequence (most common nucleotide found at each position of the alignment) of 59 PRRSV-2 strains	The recombinant virus PRRSV-CON confers significantly broader levels of heterologous protection than wild-type PRRSV [141]
VR2385	GP5: de-optimize of codon pairs	The virus, SAVE5, has reduced replicative capacity and caused significantly lower viremia and reduced lung lesions [142]. The attenuated virus could effectively protect pigs from homologous PRRSV challenges [143]
HV	nsp9: de-optimize of codon pairs	The recombination virus has weakened replication ability and could protect pigs against homologous and related PRRSV challenges. All immunized pigs survive without distinct clinical signs and pathological damage. Neutralization antibody titer and level of IFN-γ expressing CD8^+^ T cell are increased [144]
LMY	nsp1: de-optimize of codon pairs	The attenuated virus replicates slowly. The level of neutralizing antibodies and IFN-γ are not different between the attenuated viruses and original PRRSV. Importantly, pigs infected with the virus exhibit significantly reduced clinical symptoms against a heterologous challenge [145]

## Data Availability

Not applicable.

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
