# Peer review of "Porcine Reproductive and Respiratory Syndrome Virus: Immune Escape and Application of Reverse Genetics in Attenuated Live Vaccine Development"

_vaccines, 2021, doi:10.3390/vaccines9050480_

Round 1

Reviewer 1 Report

Review of "Porcine Reproductive and Respiratory Syndrome Virus: Immune Escape and Application of Reverse Genetics in Attenuated Live Vaccine Development"

This is a very well organized and written manuscript that provides a nice review of current vaccination efforts to control the important swine disease, Porcine Reproductive and Respiratory Syndrome Virus (PRRSV).  I have only minor comments that are intended to clarify the meaning of some of the expressions and terms in the text.

Line 55- "deletion" not "depletion"

Line 56- instead of "more epidemic" how about "more highly transmissible"

Line 134- Instead of "The researches show" how about "Huang et al. (28) show that..."

Line 229- how about "several references show" instead of "works show"

Table 1- In the entry for Ingelvac, references 107 and 124 are cited but there is no reference 124 in your bibliography.

Lines 382-383- "The data indicate that the approach of using cytokines as an adjuvant only has references values".  It is unclear to me what the authors mean by reference values.  Do you mean that cytokine adjuvants have no effect?

In section 5, Concluding remarks, I thought the authors might suggest that the PRRSV vaccine field consider combining the codon deoptimization strategy that provides for safer live vaccines (less chance of reversion to virulence) with the PRRSV-CON approach that appears to provide broader protection from heterologous virus strains.  This approach might provide a path forward for PRRSV vaccine research.  Just my opinion.

Reviewer 2 Report

The authors have comprehensively summed up the mechanisms of immune escape by PRRSV and have summarized different reverse genetics platforms for PRRSV constructed and used for vaccine development by different groups around the globe. The manuscript is well written and the description and figure elaborating the modulation of the Type I IFN signal pathway by PRRSV proteins is quite informative. 

General comments:

  1. The main problem with this review is that the authors have gathered all the information and jotted it down in the review. All this information is already available in the reviews previously written. In my view, the review should include the key in the knowledge about the immune escape and how they can be filled. Likewise, there should be information and ideas regarding how we can improve the reverse genetics to improve the vaccines against the PRRSV.
  2. The review is supposed to be up to date but that’s not the case here. For example, in page 2 line 46 the authors still divide PRRSV as PRRSV-1 and PRRSV-2. However, according to ICTV 2018 the nomenclature has changed.
  3. References are missing at some places which should be included e.g in Page 2 Line 63, 72 etc.
  4. In Page 2, Line 73 the authors have mentioned that the virus can’t be detected in the lungs after 4-5 weeks. Is it so? Please check again and include the reference.
  5. In page 5 Line 176, the authors on the basis of one study are generalizing that the weak adaptive immune response to PRRSV is due to the thymus dysfunction. However, the authors have not taken into consideration that the adaptive response is weak even in the mild PRRSV strains which do not affect the thymus. Moreover, the authors have not taken into consideration the differences between the local and peripheral adaptive responses.
  6. Again, the authors have generalized that the Tregs increase in the PRRSV infection. However, there are reports where the Tregs have not been reported to increase.
  7. The reverse genetics platforms summarized by the authors have already been reviewed comprehensively in a recent review paper by Chaudhari et.al. 2020 along with the applications of these platforms. The authors need to improve the review and make it more interesting to the reader.

Round 2

Reviewer 2 Report

The authors have improved the review quite well but before acceptance authors need to make more changes as stated below:

  1. In the last paragraph of section 2.1. Suppression of type I Interferon production and signaling, it would be great if authors give some ideas regarding how we can deal with or avoid the suppression of Type I IFN caused by PRRSV including mechanisms other than reverse genetics which have already been stated in section 4.1.
  2. Similarly, section 2.2. Dysregulation of cellular immune responses looks incomplete without a concluding paragraph stating how we can improve the cellular immune responses to combat PRRSV.
  3. If possible, the authors should include a diagrammatic representation of the reverse genetics systems showing some of their critical features or at least the representation of the reverse genetics system being used in the author's laboratory.
  4. There are some grammatical mistakes in the manuscript which need some attention. 
